# Serotonin Transporter Gene Polymorphisms Predict Adherence to Weight Loss Programs Independently of Obesity-Related Genes

**DOI:** 10.3390/nu17061094

**Published:** 2025-03-20

**Authors:** Mana Yatsuda, Miyako Furou, Keiko Kamachi, Kaori Sakamoto, Kumiko Shoji, Osamu Ishihara, Yasuo Kagawa

**Affiliations:** 1Nutrition Clinic, Kagawa Nutrition University, 3-24-3 Komagome, Toshima, Tokyo 170-8481, Japan; tp.m.yatsuda@eiyo.ac.jp (M.Y.); kamachi@eiyo.ac.jp (K.K.); ishihara.osamu@eiyo.ac.jp (O.I.); 2Institute of Nutrition Sciences, Kagawa Nutrition University, 3-9-21 Chiyoda, Sakado 350-0288, Japan; miyakoflowerforest@gmail.com (M.F.); sakamoto@eiyo.ac.jp (K.S.); shoji.kumiko@eiyo.ac.jp (K.S.); 3Faculty of Nutrition, Kagawa Nutrition University, 3-9-21 Chiyoda, Sakado 350-0288, Japan

**Keywords:** serotonin transporter, obesity, adherence, personality

## Abstract

Background/Objectives: Adherence to treatment instructions is essential in managing chronic diseases related to obesity. One gene associated with adherence is the serotonin transporter (5-HTTLPR) gene, which has long (L) and short (S) alleles, resulting in LL, SL, and SS genotypes. Risk alleles for obesity include the R variant of the β3-adrenergic receptor (*β3AR*) and the G variant of uncoupling protein 1 (*UCP1*). This study aimed to evaluate whether the S/L variant of *5-HTTLPR*, the R variant of *β3AR*, and the G variant of *UCP1* are associated with adherence to a weight loss program. To assess the factors influencing adherence, eating behavior was evaluated using the Eating Behavior Questionnaire (EBQ). Methods: This study included 56 well-educated and middle-class women with a mean age of 57.3 ± 10 years and a mean BMI of 27.2 ± 5.6 kg/m^2^. Long-read sequencing was used to analyze S/L mutations. Participants followed a six-month diet and exercise regimen for obesity management. Outcomes were assessed using clinical data and EBQ scores. Adherence was objectively measured by the reduction in body fat percentage. Results: Participants were classified as SS (69.6%), SL (17.9%), or LL (12.5%). The R variant of *β3AR* was present in 34% of participants, with the G variant of *UCP1* in 75%. After the intervention, SS participants showed significantly greater reductions in weight and body fat percentage than LL participants (*p* < 0.05). Among EBQ items, significant improvements (*p* < 0.05) were observed in SS participants for eating as a diversion, feeling of fullness, bad eating habits, unsteady eating patterns, and total EBQ score. In SL participants, only bad eating habits improved, whereas no significant changes were observed in LL participants. Obesity risk alleles did not significantly affect clinical outcomes, though there may be small number bias. Conclusions: SS genotype participants demonstrated higher adherence to the weight loss program, leading to improved clinical outcomes and EBQ scores, independent of obesity risk genes.

## 1. Introduction

### 1.1. Adherence and Serotonin Transporter Gene

Patient adherence is defined as the degree to which a patient follows treatment recommendations prescribed by a physician or registered dietitian. Adherence is important in the treatment of obesity-related diseases, such as metabolic syndrome, hypertension, type 2 diabetes mellitus, and cardiovascular disease [1].

Behavioral therapy, with high adherence rates, is an important component of lifestyle modification, assuming that obesity-leading behaviors have a strong educational component [1]. Stricter adherence to the Japanese Dietary Guidelines has been shown to reduce all-cause mortality (hazard ratio 0.85, *p* < 0.001) and cardiovascular disease mortality in Japanese adults [2]. The World Health Organization conceptually defines adherence as “the degree to which an individual’s behavior is consistent with agreed-upon recommendations from a health care provider” [3]. However, there is no consensus on its operational definition, as there is no universally accepted “gold standard” adherence measure or operational definition [3]. Therefore, we define objective measures as the means of data collection, rather than subjective measures such as self-reporting. Among the many objective measures, change in body fat percentage is more accurate than change in BMI. In addition to the genetic background, among the three measured proxies of socioeconomic status—education, income, and area deprivation—low education emerged as the strongest factor associated with lower adherence to a healthy diet [4]. However, in this study, the participants are well educated, middle-aged, middle-class women who can afford to pay for tuition, and there is little socioeconomic bias.

The serotonin transporter gene is associated with many human behaviors [5]. Thus, one of the authors (Y.K.) reported that adherence to weight loss advice in obese patients is influenced by polymorphisms in the promoter region of the serotonin (5-HT) transporter (*5-HTTLPR*) [6]. The serotonin transporter (5-HTT) is encoded by a gene (*SLC6A4*) on chromosome l7q11.2 [7]. The two most common alleles are the long (L: 16 repeat insertion) and short (S: 14 repeat insertion) variants [7]. The long variant of *5-HTTLPR* has higher transcriptional activity than the short variant [7]. Three single nucleotide polymorphisms in *TPH2* (brain tryptophan hydroxylase), *SLC6A4* (serotonin transporter), and *HTR3B* (serotonin receptor) were found to be significantly associated with obesity in humans [8]. The S variant of *5-HTTLPR* is associated with trait anxiety [9]. According to Cloninger’s proposal, personality is composed of temperament and character [10]. Temperament is defined in terms of perception-based habits controlled by the amygdala and other parts of the limbic system, with responses being genetically determined [10]. In contrast, character is defined in terms of concept-based values controlled by the cerebral cortex determined by learning via the hippocampus [10]. Four dimensions of temperament, namely novelty seeking, harm avoidance, reward dependence, and persistence, were added later and are independently heritable [11]. A Five-Factor Model of personality has been proposed with genetic analysis of extraversion, neuroticism, agreeableness, conscientiousness, and openness [12]. A population-based cohort study based on data from the UK Biobank found that, of the five personality traits, diligence (conscientiousness) was significantly associated with a reduced risk of cardiovascular diseases and type 2 diabetes mellitus [13]. A meta-analysis of genome-wide association studies identified six genetic loci of the Big Five Inventory [14]. However, Manhattan plots indicate that the gene locus for diligence or conscientiousness is on chromosome 7 [14] and not the *5-HTTPR* locus on chromosome 17 [7]. Indeed, no significant effect of the serotonin transporter genotypes was observed on any of the dimensions of the Big Five Inventory, NEOFFI [15]. Using positron emission tomography of ^11^C-labelled serotonin analogs, a significant positive correlation was found between self-direction and the availability of 5-HTT, suggesting that a self-confident character may be associated with enhanced serotonin neurotransmission in the L type [16]. Psychological differences in harm avoidance between S and L types were also detected by functional magnetic resonance imaging of the amygdala nucleus [17].

### 1.2. Ethnic Differences in Genes and Obesity

There are three genotypes (LL, SL, and SS) that are combinations of a long L and a short S [5,17]. Recently, 14 allelic variants have been discovered, but the combined proportion of variants with 15, 19, 20, and 22 repeats were only 3.1% in Japanese participants [18]. Significant ethnic differences in the distribution of genotypes were observed between Japanese and Caucasians: Japanese, SS = 61.9%, SL = 31.4%, LL = 0.8%, others = 6.4%; Caucasian, SS = 20.3%, SL = 48.6%, LL = 31.1%, others = 0.0% [18]. Furthermore, there are large ethnic differences in nutritional intake and obesity rates (BMI > 30 kg/m^2^, 4.8% for Japanese men and 3.7% for Japanese women, 35.5% for American men and 37.0% for American women) [19]. A detailed analysis of serotonin metabolism and obesity in black adolescents [8] showed significant difference from those in Japanese adolescents [18]. Therefore, special consideration should be given to nutritional studies of the serotonin transporter in Japanese people [19].

### 1.3. Serotonin Transporter and Energy Metabolism

Serotonin is present in all species, regulating diverse behavioral and physiological processes to maintain energy balance [20]. Serotonin neurons in the brain are essential for thermoregulation and control the metabolic activity of thermogenic fat [21]. Ablation of serotonin neurons by diphtheria toxin causes loss of thermoregulation by decreasing uncoupling protein 1 (*UCP1*) in adipocytes [21]. Energy expenditure in adults with the SS genotype was lower than that of adults with the SL and LL genotypes, and as a result, the BMI of adults with the SS genotype (26.7 ± 0.2 kg/m^2^) was higher than that of adults with the SL (26 ± 0.1 kg/m^2^) and LL (25.4 ± 0.2 kg/m^2^) genotypes (*p* < 0.0002) [22]. Genetic risk factors for obesity include both psychology-related gene polymorphisms including the S variant of 5-HTTPLPR and metabolism-related gene polymorphisms, including the R variant of β3-adrenergic receptor (*β3AR*, rs4994, W64R) [23,24] and the G variant of uncoupling protein 1 (*UCP1*, rs1800592, A3826G) [25,26].

Adipocyte lipolysis by hormone-sensitive lipase is activated by phosphorylation by cyclic AMP-dependent protein kinase A, which is activated by *β3AR* upon catecholamine binding [27]. A prospective cohort study of 373,026 individuals with genetic susceptibility to coronary heart disease found that those at moderate and high genetic risk were at higher risk than those at low genetic risk, using a weighted polygenic risk score calculated by summing the number of risk-increasing alleles at 300 single nucleotide polymorphisms and multiplying them by the corresponding effect estimates [28]. A cross-sectional study of physical activity, television viewing, and sleep duration in 233,110 adults from the UK Biobank found that behavior has a strong influence on cardiovascular disease and type 2 diabetes mellitus [29]. Therefore, to interpret the effects of adherence, it is necessary to assess participant behavior.

### 1.4. Objective of This Study

In this study, the effect of *5-HTTLPR* polymorphisms on adherence to dietary and exercise instructions for weight loss, as well as on eating behaviors that facilitate adherence, was investigated. Since obesity is influenced by both risk genes (*β3AR* and *UCP1*) and eating behaviors, the impact of these risk genes was also examined. In the previous report [6], we examined the genetic influence on adherence only by looking at improvements in blood glucose levels, but this time we took a closer look at the genetic influence on improvement of eating behavior using a number of objective indicators such as weight, BMI, and body fat percentage, as well as questionnaires. In previous studies [6], only the S and L types of serotonin transporters were known, but in this study many mutations were reported [18], though we confirmed that their frequency is so rare that their effects can be ignored. This study was unique in that the adherence patterns of Japanese individuals differ from those of other ethnic groups [7,8,9,10,11,12] in genetic [18], nutritional [2,19], anthropometric [6], and psychological [5,15] aspects. One of the authors (Y.K.) previously reported that the SS type of *5-HTTLPR* is associated with improvements in fasting blood glucose levels [6]. In a study of 264 patients (excluding those taking medication for diabetes mellitus, hypercholesterolemia, and hypertension), the changes in fasting blood glucose levels were examined after 11 weeks (short term) and 23 years (long term) of nutrition and exercise guidance [6]. Fasting blood glucose levels were significantly lower in the SS group than in the LL + SL group (*p* = 0.01 in the short term, *p* < 0.0001 in the long term) [6]. In the present study, a detailed analysis of anthropometric and blood biochemistry data was conducted to evaluate the effects of *5-HTTLPR*, *β3AR*, and *UCP1* genotypes on adherence to recommendations from the Healthy Diet Course. To enhance adherence to the course, it is essential that participants clearly understand health information related to their condition [30]. Therefore, at the beginning of the course, participants were informed of their *β3AR* and *UCP1* risk alleles but not their SS, SL, or LL genotypes. Since hypothalamic serotonin has been reported to regulate eating behavior and body weight [31], differences in eating behaviors were also analyzed using the Eating Behavior Questionnaire (EBQ) to explore the potential mechanisms underlying genotype-specific variations in anthropometric measurements and blood biochemistry.

## 2. Materials and Methods

### 2.1. Participants

The participants were 56 Japanese women with a mean age of 57.3 ± 10 years and BMI of 27.2 ± 5.6 kg/m^2^, who are well educated, middle class, and have participated in lifestyle modification programs at the nutrition clinic of the Kagawa Nutrition University, Japan. Relevant clinical information of the participants is summarized in Table 1. The participants voluntarily participated in the weight loss program for approximately 6 months between 2014 and 2017, receiving eight dietary and exercise instruction sessions, a basic knowledge course on lifestyle-related diseases, a 500 kcal lunch, and individual nutritional counseling. The items included anthropometric data (height, weight, body fat percentage, waist circumference, blood pressure) and fasting blood biochemistry. These measurements were performed before and after the intervention, and a questionnaire survey on eating behavior was also completed. Activities of daily living were within the normal range, and all participants were able to attend the clinic without assistance. Results are presented as mean ± standard deviation (SD) values. Because the subjects in our nutrition clinic were overwhelmingly female and foreign nationals were extremely rare, we relied on the literature for comparisons between men and women and between races. The experimental plan was approved by the Human Genome Medical Ethical Committee of Kagawa Nutrition University (No. 301-G, 29 May 2014, and revised No. 257-G, 10 October 2015) and conformed to the World Medical Association Declaration of Helsinki (1964, 2000 edition). Prior to the experiment, the purpose and content of this study were fully explained to the participants, and written, informed consent was obtained.

### 2.2. Dietary and Exercise Advice

The participants took the Kagawa Nutrition University Healthy Diet Course. This course consisted of both dietary and exercise instruction. Our dietary guidance is based on the “Four-Food-Group Point Method” [32], which restricts energy intake without reducing protein intake to avoid gluconeogenesis and weight rebound. To reduce their body weight without sarcopenia, the participants were instructed to consume a mixture of foods belonging to four major food groups as follows: 240 kcal each of group 1 (milk, dairy products, and egg), group 2 (protein sources: meat, fish, and beans), and group 3 (mineral, vitamin, and fiber sources: vegetables and fruits), and then their total energy intake was adjusted by varying their intake of group 4 (energy source: grains, oil, and sugar) [32]. More than 1000 patients have participated in the Healthy Diet Course in our clinic and demonstrated excellent long-term weight control [33]. In this study, the Healthy Diet Course lasted 6 months, during which diet and exercise advice was repeated 8 times. Ten typical 500 kcal meals were provided during the course. At the start of the course, health literacy education was provided, including the importance of weight management by daily weight recording and information about risk alleles of *β3AR* and *UCP1*.

### 2.3. Genotyping

For genetic testing to identify polymorphisms of *5-HTTLPR*, *β3AR*, and *UCP1*, blood was collected from the median cubital vein under fasting conditions with the consent of the participants. DNA was purified using an automated magnetic particle DNA extraction device (Magtration System 6GC; Precision System Science, Chiba, Japan) [34]. Long-read sequencing was performed at Takara Bio Inc. (Nojihigashi 7-4-38, Kusatsu, Shiga, Japan) according to the methods reported [35,36]. The promoter region of serotonin transporter gene (*5-HTTLPR*) was located in the first exon in the long arm of chromosome 17, as reported [35]. Participants were classified into SS, SL, and LL types of *5-HTTLPR*, and the measured items of each group were compared. Obesity-related gene polymorphisms of *β3AR* and *UCP1* were genotyped using an automated polymorphism analyzer by Kagawa et al. [37] after isolating DNA [34].

### 2.4. Anthropometry

Height was measured to the nearest 0.1 cm using a digital height gauge (A&D AD-6227, Tokyo, Japan), and weight was measured to the nearest 0.1 kg using a digital scale (TANITA DC-430A, Tokyo, Japan), with 0.6 kg subtracted to account for clothing weight, and the data were used for the analysis. Waist circumference was measured to the first decimal place at the umbilicus. Body fat percentage was measured to the nearest 0.1% using the DEX method (Discovery QDR W type, manufactured by Toyo Medic Co., Ltd. Kawagoe City, Japan). The weight loss rate was calculated using the following formula:Weight loss rate (%) = (Initial Weight − Final Weight)/(Initial Weight) × 100

Initial Weight: the starting weight before the intervention.Final Weight: the weight after the intervention.

The methods for calculating body fat percentage loss rates and waist circumference reduction rates are the same. Blood pressure was measured using a fully automatic blood pressure monitor (A&D automatic blood pressure monitor TM-2655P, AKTIO Corporation, Tokyo, Japan). The risk of bias due to expectation effect is rare because it is measured not by subjective adherence but by objective improvements in body fat percentage, as in this study.

### 2.5. Blood Biochemistry

The blood biochemistry values used in the analysis were obtained by drawing blood from the median cubital vein early in the morning, separating the serum, and outsourcing the analysis to LSI Medience Corporation. The blood data included fasting blood glucose, hemoglobin A1c (hereafter referred to as HbA1c), triglycerides (TGs), LDL cholesterol (LDL-C) (direct method), and total cholesterol (TC). The methods used for these biochemical analyses are all described in the “National Health and Nutrition Survey Japan, 2019 Edition”.

### 2.6. Eating Behavior Questionnaire

Eating behavior was assessed using the Eating Behavior Questionnaire (EBQ) recommended by the Japan Society for the Study of Obesity. Eating behavior was assessed by aggregating the scores of each item into seven categories: (1) perception of constitution and weight, (2) motivation for eating, (3) eating as diversion, (4) feeling of fullness and hunger, (5) bad eating habits, (6) contents of meals, and (7) unsteady eating pattern [38]. The questionnaire consisted of 66 items, and participants were asked to respond to each item on a scale of “not at all like me” (1 point), “sometimes like me” (2 points), “tendency to be like me” (3 points), and “very like me” (4 points) [38]. The subtotal scores of all categories were combined into a total score. Higher scores in each category and the total score indicate poorer eating behaviors. An octagonal diagram was created by expressing the scores in each category as a percentage of the 100% out of 4 points [38,39]. The EBQ was administered before and after the Healthy Diet Course.

### 2.7. Statistical Analysis

The Shapiro–Wilk test was used to confirm the normality of the distributions of age, number of classes, physical findings, and eating behavior scores, and the Leven test was used to confirm homogeneity of variance. The significance level was set at 5% using IBM SPSS v30. Normally distributed data are shown as mean ± standard deviation values, and non-normally distributed data are shown as median (25th–75th percentiles) values. Types of obesity-related gene polymorphisms are shown as numbers and percentages. To compare before and after the intervention, paired t-tests were used for normally distributed data, and the Wilcoxon signed rank-sum test was used for non-normally distributed data. Comparisons among genotype groups were conducted using the Kruskal–Wallis test, followed by adjustments of significance probabilities for multiple comparisons using the Bonferroni correction. To compare eating behavior scores before and after the intervention, paired *t*-tests were used for normally distributed data, and the Wilcoxon signed-rank test was used for non-normally distributed data.

## 3. Results

### 3.1. Genotypes

Genotyping showed that, of a total of 56 participants, 39 (69.6%) were SS, 10 (17.9%) were SL, and 7 (12.5%) were LL. Allele frequencies were within the Hardy–Weinberg equilibrium range. The β3-adrenergic receptor (*β3AR*, rs4994, W64R) polymorphisms were WW (66%), WR (29%), and RR (5%), and the uncoupling protein 1 (*UCP1*, rs1800592, A3826G) polymorphisms were AA (25%), AG (43%), and GG (32%). The distribution of *β3AR* genotypes among the SS, SL, and LL genotypes was similar to the overall distribution. However, the distribution of the GG genotype of *UCP1* in SL and LL was 60% and 57%, respectively.

### 3.2. Anthropometry and Blood Biochemistry

Table 2 shows the changes in the measured values of all participants before and after the intervention. After the intervention, body weight, BMI, body fat percentage, waist circumference, systolic and diastolic blood pressures, and HbA1c were all significantly decreased (Figure 1). However, the reductions in triglycerides, total cholesterol, LDL cholesterol, and fasting blood glucose were not significant. When the participants were divided into LL type, SL type, and SS type, the SS type showed a significant decrease (all *p* < 0.001) in the above measured values; however, the LL type showed no significant change (Table 3). Furthermore, multiple comparisons were performed on body weight and body fat percentage loss rates among each gene polymorphism of the serotonin transporters LL, SL, and SS. The SS type had a significantly higher reduction rate in body weight (*p* = 0.006) and body fat percentage (*p* = 0.0017) than the LL type (Figure 2).

Because the numbers of LL and SL types were significantly smaller than those of the SS type, the reduction rates of body weight and body fat percentage in the LL + SL type were compared with those of the SS type (Figure 3). Again, the SS type had significantly higher reduction rates of both body weight (*p* = 0.006) and body fat percentage (*p* = 0.018) than the LL + SL type (Figure 3). When comparing the reduction rates of waist circumference, the SS type showed no significant difference compared to the SL and LL types after conducting the Kruskal–Wallis test with adjustments by the Bonferroni correction (Figure 4). However, in the two-group comparison, the SS type showed a significantly higher reduction rate than the LL + SL type (*p* = 0.007) (Figure 5).

The effects of polymorphisms in *β3AR* (WW, WR, and RR) on body weight (*p* = 0.568), body fat percentage (*p* = 0.246), and waist circumference were all insignificant. The effects of polymorphism in *UCP1* (AA, AG, and GG) on the change in those anthropometric data were also insignificant, except for the change in waist circumference of AA vs. AG (*p* = 0.05). The effects of the combinations of SS, SL, and LL and *β3AR* (WW, WR, and RR) and *UCP1* (AA, AG, and GG) on the change in body weight, body fat percentage, and waist circumference were all insignificant.

### 3.3. Eating Behavior Scores

Comparison of the eating behavior scores of the SS, SL, and LL participants is summarized in Figure 6. In the SS group, the items that improved significantly after the intervention were eating as diversion, feeling of fullness, bad eating habits, unsteady eating pattern, and overall (Figure 6, right). In contrast, in the SS group, there was no improvement in perception of constitution, motivation of eating, and contents of meals (Figure 6, right). Only bad eating habits improved significantly in the SL group (Figure 6, middle), and no items improved in the LL group (Figure 6, left). Because the number of participants of the SL and LL types was smaller than that of the SS type, the LL and SL types were grouped together (Figure 7 and Table 4). In the LL + SL group, none of the items improved (Figure 7, left) (Table 4). In addition, the tested risk gene variants of *β3AR* and *UCP1*, of which the participants were informed, led to good adherence to diet and exercise advice.

## 4. Discussion

It has been shown that adherence to the instructions required for the prevention of chronic diseases is influenced by genetic polymorphisms that affect individual personality. In this study, adherence is operationally defined by objective improvements in body fat percentage and is limited to SS type (Table 3 and Figure 2). The genetic regulation of adherence to advice on a healthy eating course by the serotonin pathway of the *5-HTTLPR* in response to dietary behavior was demonstrated. It was found that female carriers of the SS type of the *5-HTTLPR* had significant reductions in weight (Figure 2 and Figure 3), BMI (Table 3), body fat percentage (Table 3), waist circumference (Figure 5), systolic and diastolic blood pressures, and HbA1c (Table 3). In contrast, both LL and SL carriers showed significantly lower decreases in the above measures throughout the program (Figure 3 and Figure 5). The improvement in clinical data for SS participants may be due to behavioral improvements in SS participants (Table 4, Figure 6 and Figure 7), that is, SS participants showed better adherence in both clinical data (Table 3, Figure 2 and Figure 3) and feeding behavior (Table 4, Figure 6 and Figure 7), whereas SL and LL participants did not. In a preliminary study report by one of the authors (Y.K.) that did not include diabetic patients, after advice on a healthy diet course [6], an association with a decrease in fasting blood glucose levels was observed only in the SS group, but data on clinical data other than blood glucose levels and on eating behavior were lacking. In the present study that included diabetic patients, no significant decrease in fasting blood glucose levels was observed, except in the SL group, due to the large fluctuations in blood glucose levels in diabetic patients (Table 3). However, the long-term average of blood glucose reflected by HbA1c was significantly decreased in the SS type (Table 3). This study is unique in that the background of medication adherence in Japanese people is different from other ethnicities [7,8,9,10,11,12] in terms of genetic [18], nutritional [2,19], anthropometric [6], and psychological [5,15,40] aspects. Significant ethnic differences in genetics were observed between Japanese and Caucasian populations in the distribution of the SS genotype of *5-HTTLPR*, 61.9% in Japanese and 20.3% in Caucasian populations [18]. The distribution of the GG genotype of UCP1 was exceptionally high between SL and LL, 60% and 57%, respectively, possibly due to natural selection due to high energy expenditure caused by serotonin-induced UCP1 activity [21]. In terms of nutrition, there are large differences in nutritional intake [19], and as a result, there are large ethnic differences in the obesity rate (BMI > 30 kg/m^2^, 3.7% in Japanese women and 37.0% in American women) [19]. In terms of anthropometric aspects, the SS type of the *5-HTTLPR* polymorphism has been reported as a risk gene for obesity in Caucasians [22] and people of African descent [8], but in the Japanese participants of this study, there were no significant differences in weight, BMI, or waist circumference between the polymorphisms. In the cohort of South American origin studied, the BMI (30.0 ± 0.2 kg/m^2^) [8] was higher than that of the Japanese participants in the present study (27.2 ± 5.6 kg/m^2^) (Table 1). In terms of psychological aspects, the NEO five-factor inventory scores of Japanese people [15] differed from those of Americans [40]. In terms of metabolic aspects, the energy expenditure of adults with the SS genotype was lower than that of adults with the SL and LL genotypes; therefore, the BMI of adults with the SS genotype was higher than that of adults with the SL and LL genotypes (*p* < 0.0002) [22]. However, the fact that SS people are more likely to follow medical advice does not mean that they are in better health, because there were significantly more centenarians with the LL genotype than with the SS genotype [15]. In fact, persons with the SS type have the tendency to overeat called disinhibition [41]. The S carriers showed an impairment to lose weight, had a lower inhibition capacity, and showed more failure (1.6 times) to control the amount of food eaten than the LL type (*p* < 0.05) [41]. Associations between Five-Factor Model personality traits [12,13,14], dietary patterns, and BMI have been reported [42]. High scores on the health aware diet dimension were associated with high conscientiousness [42], as supported by genetic analysis of the fifth chromosome [13]. To elucidate these effects, detailed analyses of brain physiology and genetics are needed. Positron emission tomography with an 18F-labelled serotonin analogue in midbrain raphe nuclei showed that a proportionate reduction in serotonin transporter density may be associated with adherence to taking antidepressants [43].

### Strength and Limitations of This Study

One strength of this study is its detailed analysis of adherence based on the polymorphism of the serotonin transporter gene, incorporating biochemical, anthropometric, and behavioral data collected before and after precise instructions were given. The long-lasting effect of the adherence was shown in the previous article [6]; the effect of improving blood glucose levels after nutritional guidance was confirmed for an average of 23 years after the course. In reference [33], subjects taught in their 30s and 40s were followed up for 68.87–7.74 years, and the effect of nutritional guidance was found to be persistent.

However, this study was limited to Japanese educated middle-class women, whose genetic background differs significantly from that of other ethnic groups [18,44] due to the long evolutionary history of adaptation to distinct environments [45]. Adherence is known to be influenced by complex factors, including genetic as well as socioeconomic factors [4], especially in other societies. Though adherence is partly influenced by serotonin transporter polymorphisms in the brain, this study lacked brain imaging data to support this connection [16,17,43]. The serotonergic nuclei, located in the midbrain raphe nucleus, project to a wide area of the brain, regulating the sympathetic nervous system and influencing feeding behavior [46]. Another limitation of this study is that all participants were female. Notably, sex differences exist in the association between serotonin transporter gene polymorphisms and personality traits [47]. This difference may be attributed to the higher methylation of the CpG island of the *5-HTTLPR* gene in S-type females compared with males (*p* < 0.0008), leading to lower mRNA production (*p* < 0.0001) [48]. This epigenetic effect is specific to SS-type females, which may contribute to increased variability in clinical and behavioral data. Nevertheless, SS-type individuals still exhibit the highest adherence to instructions. In future precision nutrition, comprehensive genetic analysis with artificial intelligence will be applied to improve adherence.

## 5. Conclusions

In this report, adherence was objectively measured by the reduction in body fat percentage. Despite being at risk for obesity, SS-type *5-HTTLPR* participants may have demonstrated better adherence to instructions than SL-type and LL-type *5-HTTLPR* participants. As a result, they exhibited significant improvements in eating behavior, blood biochemistry, and anthropometric measurements. In the future, it will be essential to refine instructional methods to enhance adherence in LL-type participants and to investigate the genetic and epigenetic mechanisms underlying adherence. This study will contribute to future precision medicine.

## Figures and Tables

**Figure 1 nutrients-17-01094-f001:**
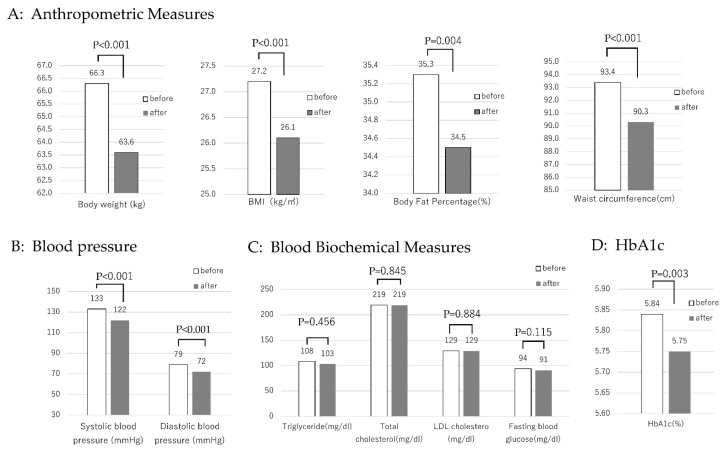
Changes in anthropometric measurements and fasting blood chemistry before and after the intervention (total, n = 56). Values are means. Anthropometric measurements showed a significant decrease (**A**). After the intervention, systolic and diastolic blood pressure decreased significantly (**B**). Blood chemistry results did not decrease significantly (**C**). After the intervention, HbA1c decreased significantly (**D**).

**Figure 2 nutrients-17-01094-f002:**
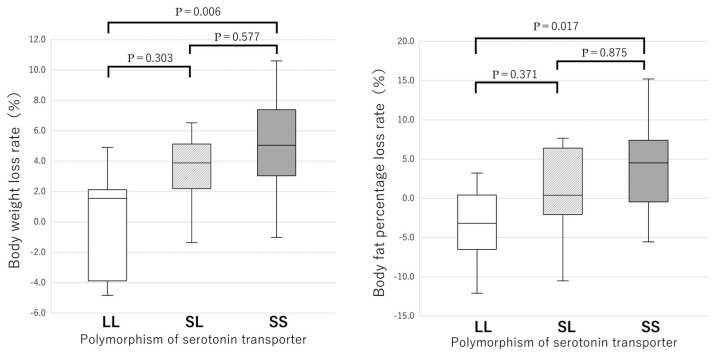
Body weight loss and body fat percentage loss rate (%) for each gene polymorphism of the promoter of serotonin transporter. Results of the boxplot are expressed median (25th–75th percentiles) values. Regarding the weight loss rate, the results of the Kruskal–Wallis test for independent samples revealed the significance probability adjusted by the Bonferroni correction.

**Figure 3 nutrients-17-01094-f003:**
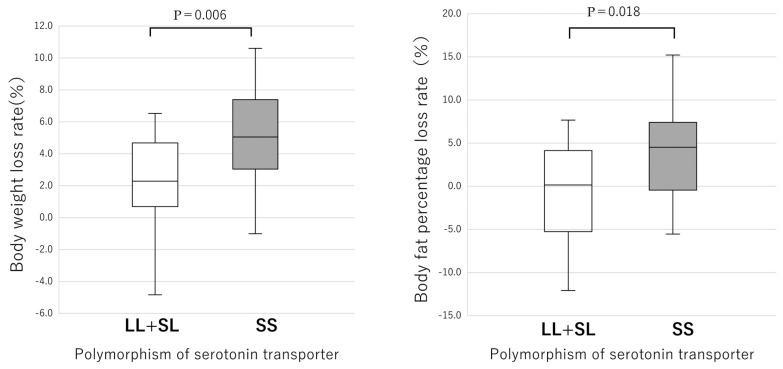
Body weight loss and body fat percentage loss rate (%) for LL + SL and SS genotypes of the promoter of serotonin transporter. The LL and SL types, which had a small sample size, were combined and compared with the SS genotype. Results are expressed as median (25th–75th percentiles) values. The results of the Mann–Whitney U test for independent samples indicate a significant difference.

**Figure 4 nutrients-17-01094-f004:**
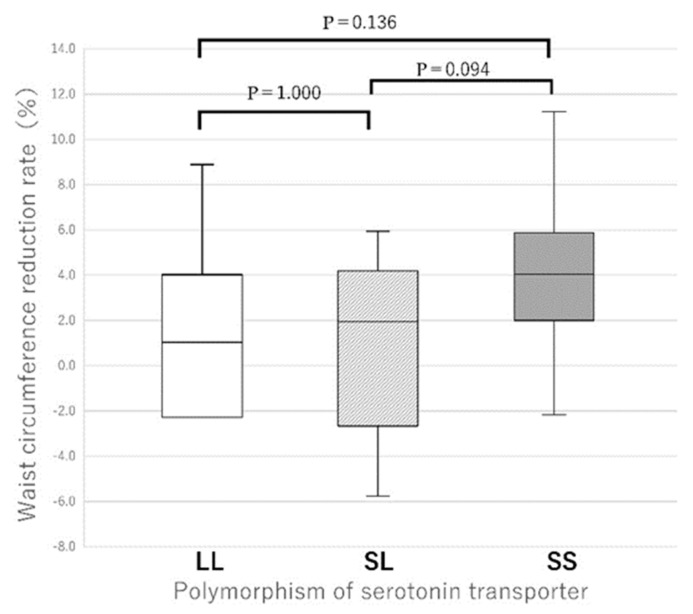
Waist circumference reduction rates for each gene polymorphism of the promoter of serotonin transporter. Results of the boxplot are expressed median (25th–75th percentiles) values. The Kruskal–Wallis test for independent samples was conducted, and subsequent adjustments by the Bonferroni correction showed no significant differences.

**Figure 5 nutrients-17-01094-f005:**
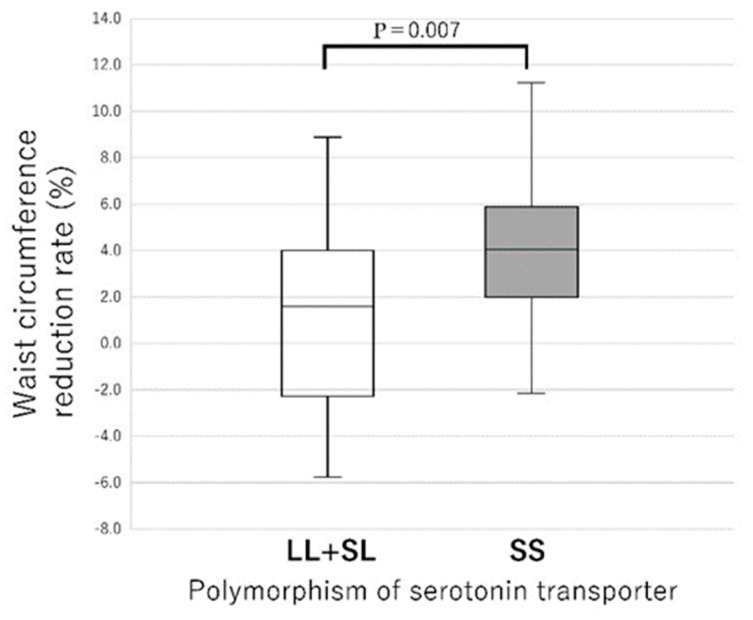
Waist circumference reduction rates for each gene polymorphism of the promoter of serotonin transporter. Results are expressed as median (25th–75th percentiles) values. The results of the Mann–Whitney U test for independent samples indicate a significant difference.

**Figure 6 nutrients-17-01094-f006:**
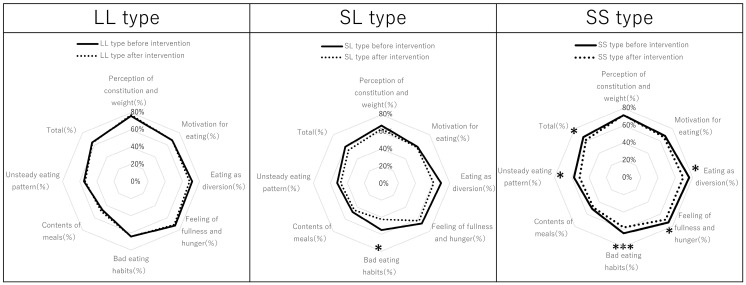
Comparison of the octagon diagrams plotting eating behavior of participants before and after the Healthy Diet Course according to the promoter of serotonin transporter gene polymorphisms of SS, SL, and LL types. The results are comparisons before and after the intervention within each group. The Wilcoxon signed-rank test for paired samples was conducted, indicating a significant difference. * *p* < 0.05, *** *p* < 0.001.

**Figure 7 nutrients-17-01094-f007:**
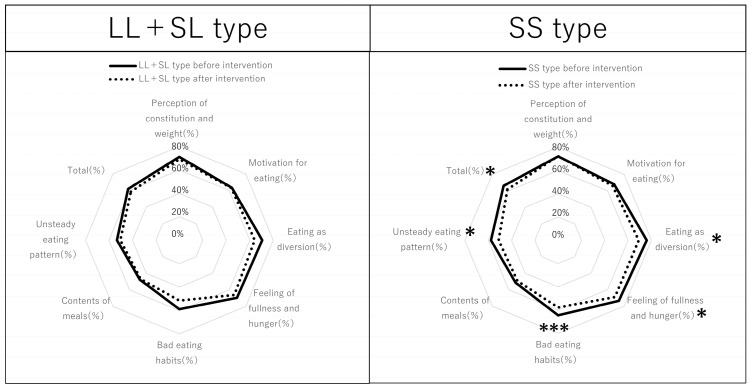
Comparison of the octagon diagrams plotting eating behavior in participants before and after the Healthy Diet Course according to promoter of serotonin transporter gene polymorphisms of the SS and LL + SL types. The results are comparisons before and after the intervention within each group. The Wilcoxon signed-rank test for paired samples was conducted, indicating a significant difference. * *p* < 0.05, *** *p* < 0.001.

**Table 1 nutrients-17-01094-t001:** Participants’ characteristics (n = 56).

	Mean ± SD
Body weight (kg)	66.3 ± 13.8
BMI (kg/m^2^)	27.2 ± 5.6
Body fat percentage (%)	35.3 ± 6.1
Waist circumference (cm)	93.4 ± 13.9
Systolic blood pressure (mmHg)	133 ± 18
Diastolic blood pressure (mmHg)	79 ± 12
Triglycerides (mg/dL)	108 ± 65
Total cholesterol (mg/dL)	219 ± 38
LDL cholesterol (mg/dL)	129 ± 34
Fasting blood glucose (mg/dL)	94 ± 22
HbA1c (%)	5.8 ± 0.6

Abbreviations: BMI, body mass index; HbA1c, hemoglobin A1c.

**Table 2 nutrients-17-01094-t002:** Changes in measurements before and after intervention (overall, n = 56).

	Before Intervention		After Intervention		*p* Value
	Mean	±SD	Mean	±SD	
Body weight (kg)	66.3	±13.8	63.6	±13.6	<0.001
BMI (kg/m^2^)	27.2	±5.6	26.1	±5.6	<0.001
Body fat percentage (%)	35.3	±6.1	34.5	±5.8	0.004
Waist circumference (cm)	93.4	±13.9	90.3	±13.6	<0.001
Systolic blood pressure (mmHg)	133	±18	122	±16	<0.001
Diastolic blood pressure (mmHg)	79	±12	72	±11	<0.001
Triglyceride (mg/dL)	108	±65	103	±45	0.456
Total cholesterol (mg/dL)	219	±38	219	±31	0.845
LDL cholesterol (mg/dL)	129	±34	129	±27	0.884
Fasting blood glucose (mg/dL)	94	±22	91	±18	0.115
HbA1c (%)	5.84	±0.59	5.75	±0.63	0.003

Abbreviations: BMI, body mass index; HbA1c, hemoglobin A1c. Statistical significance before and after the intervention was tested using the Wilcoxon signed-rank test.

**Table 3 nutrients-17-01094-t003:** Effects of serotonin transporter gene polymorphism on weight, percentage, length, pressure and concentrations before and after intervention.

	LL Type (n = 7)			SL Type (n = 10)			SS Type (n = 39)		
	Before	After		Before	After		Before	After	
	Mean ± SD	Mean ± SD	*p* Value	Mean ± SD	Mean ± SD	*p* Value	Mean ± SD	Mean ± SD	*p* Value
Body weight (kg)	69.7 ± 19.8	70.1 ± 22.3	1.000	62.1 ± 12.7	57.2 ± 8.3	0.007	67.5 ± 13.4	64.1 ± 12.5	<0.001
BMI (kg/m^2^)	28.6 ± 7.6	28.9 ± 8.8	0.917	25.4 ± 4.1	24.5 ± 3.8	0.009	27.4 ± 5.6	26.0 ± 5.2	<0.001
Body fat percentage (%)	33.9 ± 5.6	35.0 ± 5.5	0.910	34.8 ± 6.9	34.2 ± 6.2	0.333	35.7 ± 6.0	34.4 ± 5.9	<0.001
Waist circumference (cm)	98.0 ± 15.8	97.0 ± 18.3	0.672	88.3 ± 11.4	87.0 ± 9.3	0.306	93.9 ± 14.0	90.0 ± 13.5	<0.001
Systolic blood pressure (mmHg)	131 ± 22	123 ± 23	0.116	134 ± 21	123 ± 17	0.058	133 ± 17	122 ± 15	<0.001
Diastolic blood pressure (mmHg)	76 ± 11	72 ± 16	0.128	76 ± 11	66 ± 11	0.038	80 ± 12	73 ± 9	<0.001
Triglycerides (mg/dL)	99 ± 43	93 ± 32	0.612	93 ± 54	103 ± 66	0.919	113 ± 70	104 ± 48	0.433
Total cholesterol (mg/dL)	191 ± 25	192 ± 16	0.866	222 ± 33	231 ± 38	0.906	223 ± 40	221 ± 29	0.759
LDL cholesterol (mg/dL)	107 ± 23	111 ± 15	0.446	133 ± 30	137 ± 31	0.415	132 ± 36	131 ± 27	0.794
Fasting blood glucose (mg/dL)	90 ± 9	86 ± 3	0.207	114 ± 30	101 ± 29	0.014	89 ± 16	89 ± 15	0.816
HbA1c (%)	5.6 ± 0.2	5.5 ± 0.3	0.026	6.2 ± 0.9	6.1 ± 1.1	0.356	5.8 ± 0.5	5.7 ± 0.5	0.045

Statistical significance before and after the intervention was tested using the Wilcoxon signed-rank test.

**Table 4 nutrients-17-01094-t004:** Percentage of the total possible scores in the seven categories and total of eating behavior before and after the intervention by serotonin transporter genotype.

	LL + SL Type (n = 17)			SS Type (n = 39)		
	Before	After	*p* Value	Before	After	*p* Value
	Mean	Mean		Mean	Mean	
Perception of constitution and weight (%)	71.1	69.1	0.599	72.0	72.2	0.859
Motivation for eating (%)	63.2	62.7	0.975	67.9	66.0	0.355
Eating as diversion (%)	70.6	64.2	0.078	76.1	69.1	0.040
Feeling of fullness and hunger (%)	69.4	66.2	0.180	73.6	69.0	0.010
Bad eating habits (%)	58.8	51.5	0.115	64.5	57.7	<0.001
Contents of meals (%)	47.6	46.8	0.788	51.5	49.4	0.319
Unsteady eating pattern (%)	53.2	50.3	0.248	57.6	51.0	0.001
Total (%)	61.8	58.4	0.169	66.1	61.6	0.001

Statistical significance before and after the intervention was tested using the Wilcoxon signed-rank test.

## Data Availability

The original contributions presented in this study are included in the article. Further inquiries can be directed to the corresponding author(s).

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
