# Peer review of "Serotonin Transporter Gene Polymorphisms Predict Adherence to Weight Loss Programs Independently of Obesity-Related Genes"

_nutrients, 2025, doi:10.3390/nu17061094_

Round 1

Reviewer 1 Report

Comments and Suggestions for Authors

Dear Authors

This manuscript demonstrated that Serotonin Transporter Gene Polymorphisms Predict Adherence to Weight Loss Programs Independently of Obesity-Related Genes.

Here are the major comments. 

1. This manuscript only describes the phenomena, and the biological meaning behind them is not adequately addressed. It is necessary to supplement the results of each figure in this regard

2. Why did the authors only add changes of "body weight" and "body loss" as Figure 1. Please, add all parameters in Table 2 as figures. 

3. Why did the authors not include Figures regading before and after intervention? 

    All figures regarding the significance before and after intervention were missing. 

4. In figures 3 and 5, the authors suddenly compare LL+SL and SS without enough explanation. 

The authors should explain these figures in detail. 

5. There were some inconsistencies or insufficient explanations between the Table and each Figure. For example,  in the case of LL, body weight went from 69.7±19.8 (before intervention) to 70.1±22.3 (after intervention). Body weight weight loss weight from -4.0 to +2.0
Please, check these. 

6. Please, check and correct the "kg/m2" in the abstract. 

7. Please, check the position "1.1, 1.3, 1.4, and 1.5" in the sentence. 

Author Response

Thank you for your important comments. The main problems you pointed out are corrected as follows. The sentence added both in Abstract and Conclusions is “adherence was objectively measured by the reduction of body fat percentage.” Because there is no “gold standard” of operational definition does not exist, as pointed out by Burleson, J. et al. Patient Prefer Adherence. 2025; 19: 319-344,

  1. This manuscript only describes the phenomena, and the biological meaning behind them is not adequately addressed. It is necessary to supplement the results of each figure in this regard.

      Answer: We will supplement the results of each figure as described in 2.

  1. Why did the authors only add changes of "body weight" and "body loss" as Figure 1. Please, add all parameters in Table 2 as figures.

      Answer: We added all parameters in Table 2 as figures. Since the units of parameters are different, Fig. 1 is divided into Fig. 1A, 1B, 1C and 1D.

  1. Why did the authors not include Figures regarding before and after intervention? All figures regarding the significance before and after intervention were missing.

       Answer: These are added.

  1. In figures 3 and 5, the authors suddenly compare LL+SL and SS without enough explanation. The authors should explain these figures in detail.

        Answer: Because the numbers of LL and SL are smaller than SS, so we compared LL+SL to SS. We add more  explanation to the old legends.

  1. There were some inconsistencies or insufficient explanations between the Table and each Figure. For example, in the case of LL, body weight went from 69.7±19.8 (before intervention) to 70.1±22.3 (after intervention). Body weight loss weight from -4.0 to +2.0. Please, check these.

         Answer: We have corrected, because the number of LL is only 7, and this increase of body weight was caused bias, yet the change was insignificant.

  1. Please, check and correct the "kg/m2" in the abstract.

      Answer: We have corrected.

  1. Please, check the position "1.1, 1.3, 1.4, and 1.5" in the sentence.

       Answer: In Introduction, we deleted one section, and the total number of sections are reduced to 1.1, 1.2, 1.3 and 1.4. because referee 2 recommended to reduce explanation on serotonin.

Reviewer 2 Report

Comments and Suggestions for Authors

Abstract

The relationship between genotype and adherence is not precisely formulated

Only 56 women, potential bias due to small sample size

No direct comparison with individuals without genetic analysis

Introduction
Excessive focus on serotonin: Other factors, such as socioeconomic conditions, are barely considered

Lack of context: No comparison with previous studies on genetic influence on adherence

Missing definition of key terms: Adherence and its measurement are not sufficiently described

Methods

Only Japanese women, no men or other ethnicities, limiting generalizability

Risk of bias due to expectation effects

No analysis of other factors that might influence adherence

Results

No correction for multiple testing mentioned

Some p-values are borderline and not validated through additional analyses

No information on adherence over an extended period

Discussion

Genetic influence may be overestimated

Restrictions are mentioned late and not discussed in depth

No clear statement on how the results could be applied in practice

Conclusion

Strong statements about the genetic role without sufficient empirical validation.

No clear outline of necessary futher Investigation 

Author Response

Thank you for your important comments. The main problem you pointed out is “Missing definition of key terms: Adherence and its measurement are not sufficiently described.” As described by Burleson, J. et al. Patient Prefer Adherence. 2025; 19: 319-344, conceptional definition of adherence by WHO is “the extent to which a person’s behavior corresponds with agreed recommendations from a health care provider”. However, there is no “gold standard” of operational definition does not exist. Thus, we use objective measures of adherence by reduction of body fat percentage. Only in SS genotype of 5-HTTLPR after the advice resulted in significant decrease of body fat percentage. So, in Abstract and Conclusion the following sentence was added.  “In this report, adherence was objectively measured by the reduction of body fat percentage.”

Comments and Suggestions for Authors

Abstract

  • The relationship between genotype and adherence is not precisely formulated.

      Answer: Owing to the limitation of Abstract length detailed formulation will be added in the introduction and discussion.

  • Only 56 women, potential bias due to small sample size.

     Answer: The following sentence is added. “though there may be small number bias.”

  • No direct comparison with individuals without genetic analysis

     Answer: There may be effects by other than gene, such as socioeconomic influence. Since the participants are well-educated and middle-class, socioeconomic effects may be reduced in this study, “well-educated and middle-class” is added.  

Introduction

  • Excessive focus on serotonin: Other factors, such as socioeconomic conditions, are barely considered.

    Answer: We reduced excessive comments on serotonin by reducing sections from 1.1 - 1.5 to 1.1 -1.4. We removed old references [3]and [4], instead, add the following sentences on socioeconomic conditions with a new reference [4].

“In addition to the genetic background, among the three measured proxies of socioeconomic status – education, income, and area deprivation – low education emerged as the strongest factor associated with lower adherence to a healthy diet [4]. However, in this study, the participants are well-educated, middle-aged, middle-class women who can afford to pay for tuition, and there is little socio-economic bias.”

New reference [4]

Carrasco-Marín, F.; Parra-Soto, S.; Bonpoor, J.; Phillips, N.; Talebi, A.; Petermann-Rocha, F.; Pell, J.; Ho, F.; Martínez-Maturana, N.; Celis-Morales, C.; Molina-Luque, R.; Molina-Recio, G. Adherence to dietary recommendations by socioeconomic status in the United Kingdom biobank cohort study Front Nutrition 2024; 11: 1349538.

  • Lack of context: No comparison with previous studies on genetic influence on adherence

     Answer: In the previous report, we examined the genetic influence on adherence only by looking at improvements in blood glucose levels, but this time we took a closer look at the genetic influence on improvement of eating behavior using a number of objective indicators such as weight, BMI, and body fat percentage, as well as questionnaires. In the previous studies [6], only the S and L types of serotonin transporters were known, but in this study many mutations were reported [18], but we confirmed that their frequency is so rare that their effects can be ignored.

  • Missing definition of key terms: Adherence and its measurement are not sufficiently described.

     Answer: The following text and the reference have been added in place of line 51-57:

The World Health Organization conceptually defines adherence as "the degree to which an individual's behavior is consistent with agreed-upon recommendations from a health care provider."[3]. However, there is no consensus on its operational definition, as there is no universally accepted "gold standard" adherence measure or operational definition.[3] Therefore, we define objective measures as the means of data collection, rather than subjective measures such as self-report. Among the many objective measures, change in percentage body fat is more accurate than change in BMI.

The reference added as [3].

Burleson, J.; Stephens, D.E.; Rimal, R.N. Adherence Definitions, Measurement Modalities, and Psychometric Properties in HIV, Diabetes, and Nutritional Supplementation Studies: A Scoping Review. Patient Prefer Adherence. 2025; 19: 319-344.

Methods

  • Only Japanese women, no men or other ethnicities, limiting generalizability

    Answer: Because the subjects in our nutrition clinic were overwhelmingly female and foreign nationals were extremely rare, we relied on the literature for comparisons between men and women and between races.

  • Risk of bias due to expectation effects

  Answer: The following sentence was added in the anthropometric section. “The risk of bias due to expectation effect is rare because it is measured not by subjective adherence but by objective improvements in weight and body fat percentage, as in this study.”

  • No analysis of other factors that might influence adherence.

Answer: The following sentences are added both in Introduction and Method sections. Non-genetic factors, especially education of socioeconomic status is important [4]. Participants are generally well-educated, middle-aged, middle-class women who can afford to pay for tuition, and there is little socio-economic bias among participants.

Results

  • No correction for multiple testing mentioned. Some p-values are borderline and not validated through additional analyses

  Answer: We adjusted significance probability by the Bonferroni correction after the Kruskal-Wallis test. We added the sentence in “Statistical analysis” section and each figure’s explanation.

  • No information on adherence over an extended period

   Answer: We have several follow-up data, so the following sentences are added to line 455-459.

The long-lasting effect of the adherence was shown in the previous article [6]; the effect of improving blood glucose levels after nutritional guidance was confirmed for an average of 23 years after the course. In reference [33], subjects taught in their 30s and 40s were followed up for 68.87-7.74 years and the effect of nutritional guidance was found to be persistent.

Discussion

  • Genetic influence may be overestimated

    Answer: The following sentences are added to line 462 of the original manuscript: Adherence is known to be influenced by complex factors, including genetic as well as socioeconomic factors [4] especially in other societies.

  • Restrictions are mentioned late and not discussed in depth

    Answer: The following sentences are added to line 57-62.

In addition to the genetic background, among the three measured proxies of socioeconomic status – education, income, and area deprivation – low education emerged as the strongest factor associated with lower adherence to a healthy diet [4]. However, in this study, the participants are well educated, middle-aged, middle-class women who can afford to pay for tuition, and there is little socio-economic bias.

  • No clear statement on how the results could be applied in practice

    Answer: The following sentence is added to line 473. In future precision nutrition, comprehensive genetic analysis with artificial intelligence will be applied to improve adherence.

Conclusion

  • Strong statements about the genetic role without sufficient empirical validation.

    Answer: Added “may have”

  • No clear outline of necessary further Investigation

  Answer: The following sentence will be added to line 483-484 "This study will contribute to future precision medicine".

Round 2

Reviewer 2 Report

Comments and Suggestions for Authors

None